# The Entrepreneurial Ecosystem in the Eastern Cone of Lima as a Predictor of Competitiveness and Sustainable Development

Jesús Fernando Bejarano Auqui [1,*] , Adriana Ruiz Berrio [2] , Raúl Rodríguez Antonio [3] and Beatriz Estefany Aguado [4]

1    Faculty of Business Studies, School of Management, Peruvian Union University, Lima 100, Peru
2    Faculty of Business and Law, Montemorelos University, Montemorelos Nuevo León 67530, Mexico
3    Faculty of Education, Montemorelos University, Montemorelos Nuevo León 67530, Mexico
4    Faculty of Business Sciences, Peruvian Union University, Lima 100, Peru
*    Correspondence: jesusbejarano@upeu.edu.pe; Tel.: +51-989-249-744

**Abstract:** Great changes and contradictions have occurred in the economy in recent years, forcing entrepreneurs to seek solutions to increase competitiveness, maintain growth and implement mechanisms that can create permanent solutions in the future, thus promoting development. In this context, the aim of this study is to investigate whether the perception of the entrepreneurial ecosystem is a significant predictor of the perception of competitiveness and sustainable development in the Peruvian case. For this, a structural equation model was tested using data obtained from a convenience sampling method ($n = 240$) along with a 32-item scale, which was adapted and validated using an exploratory factor analysis. Participants of this study were actors from the entrepreneurial ecosystem of the eastern cone of Lima, 79.1% of whom were entrepreneurs and businessmen (6.2% from the industrial sector, 50.5% from the commercial sector and 22.4% from the service sector); the remaining 20.9% of the sample corresponds to university professors, students, directors and advisors of incubators and accelerators. Findings of this study suggest that the perception of these actors of the entrepreneurial ecosystem has a significant effect on its perception of competitiveness and sustainable development.

**Keywords:** competitiveness; entrepreneurial ecosystem; structural equation model; sustainable development

## 1. Introduction

The study of the entrepreneurial ecosystem is attracting more and more researchers and academics from various countries on all continents. Studies such as Doing Business 2020 investigate the regulations that enhance business activity and those that constrain it. Doing Business presents quantitative indicators on business regulations and the protection of property rights that can be compared across 190 economies, World Bank (2020). Another one, Global Entrepreneurship Monitor (GEM) began in 1999 with information on the state of entrepreneurship and entrepreneurial ecosystems across the globe, and today it is carried out in 120 countries. GEM can confidently stake a claim to be the largest ongoing study of entrepreneurial dynamics in the world, GEM (2021). In addition, the other one is The Global Startup Ecosystem Index by StartupBlink, which provides two sets of rankings: the first is for countries, and the second is for individual ecosystems nestled within the cities. In 2022, this study was performed in 100 countries and 1000 cities Global-E (2022).

Other studies, such as Monitor of Dynamic and Innovative Entrepreneurship Policies in Latin America, today have 13 countries, Álvarez Martínez et al. (2022), and the study of Index of Dynamic Entrepreneurship (IDE) as an engine of sustainable development report contains the global ranking, as well as data and analyzes over 40 countries. The Index sheds light on the current imbalances and gaps between regions and countries concerning

their conditions that affect the emergence and development of dynamic and sustainable new ventures, Kantis et al. (2022).

The importance of entrepreneurship ecosystems lies in the fact that they are being considered as vehicles for entrepreneurship, Sevilla-Bernardo et al. (2022) to help small and medium-sized enterprises (SMEs) achieve their performance goals, Sharfaei et al. (2022). Creating more and better companies from the development of entrepreneurial ecosystems helps countries, regions, cities, towns, and universities to impact economic growth through innovation, competitiveness, trade, financial systems, infrastructure development, and employment, which ultimately leads to improved quality of life, competitiveness and sustainable development, Apostu et al. (2022); Calanchez Urribarri et al. (2022); Fiore et al. (2019); Kantis et al. (2022); Torres-Salazar et al. (2019).

The commitment of the actors of the entrepreneurial ecosystem plays a vital role in the search for competitiveness and sustainable development. The contribution of the universities' programs with entrepreneurial education, business incubators created to help entrepreneurs generate and evaluate business projects and ideas in their different phases. The accelerators are programs dedicated to helping entrepreneurs define their ideas following the guidelines of the Lean Start-up methodology and later provide them with tools to build their first prototypes. In addition to the other actors, such as angel investors, networks, entrepreneurship support institutions, financing institutions, mentor banks, among others, which are important for entrepreneurs to create and operate their companies, Apostu et al. (2022); Kassim et al. (2022); Mansoori et al. (2019); Sevilla-Bernardo et al. (2022).

The commitment of governments and the business environment play an important role in the development of the entrepreneurial ecosystem in a five-fold helix model: university-industry-government-public-environment, Apostu et al. (2022). Starbird et al. (2022) mention competition, trade, regulation, macroeconomic conditions, public health, and government policy as factors that impact the performance of small and medium-sized enterprises (SMEs), and therefore, it is important to align the strategy with the broader environment. Since the business environment is closely related to industry and geography, research often focuses on region or country-specific entrepreneurship ecosystems.

Achiquen Millán et al. (2021) state that relatively little research has been such as on entrepreneurship in emerging economies in developing countries, but the study is important because firms in such cases face greater resource constraints and immature markets due to their small size, which generates differences in business behavior in developed countries. In Peru, one of the strategies of the current government to help the growth of the business fabric is entrepreneurship as a dynamic factor to provide people with economic resources in order to develop productive projects.

Romero-Parra et al. (2022) argue that entrepreneurship is highly valued in Peru, and in the Latin American context it is the first country with the greatest awareness of the entrepreneurial ecosystem and the third in development in entrepreneurial skills and competencies. The need to know whether the public policy developed in Peru with the FINCyT Science and Technology Program, which began in 2006 to support the Peruvian entrepreneurial ecosystem, is positively impacting in competitiveness and sustainable development, which gives rise to the research question of this article.

The rest of the article is organized as follows: (2) Literature review, (3) Methodology, (4) Results, (5) Discussion, (6) Conclusions, (7) Recommendations, and (8) Implications.

## 2. Literature Review

### 2.1. Entrepreneurial Ecosystem

During the last two decades, various works of literature have addressed the interest of governments, private organizations, Organizaciones No Gubernamentales (ONGs) [Non-Governmental Organizations] and universities in entrepreneurship as strategic development. Guevara Gómez et al. (2022) indicated that the level of economic and social development of a country depends largely on its entrepreneurial dynamics and the joint

effort of the public and private sectors to promote the culture of entrepreneurship in people, especially in the country with high standards of innovation, investigation and development. In this regard, Comision Economica para America Latina y el Caribe CEPAL (2017) [Economic Commission for Latin America and the Caribbean] emphasizes entrepreneurship as a strategy and solution to reactivate economic growth and social progress in Latin America and the Caribbean. Entrepreneurial skills and spirit can enable young people to develop knowledge-based economic activities, improve productivity and transform the region's policies into sustainable development.

In this logic, Zaldívar Puig et al. (2019) contextualize the entrepreneurial ecosystem as the set of economic, legal, institutional, political, social, cultural and environmental elements that impact the development of enterprises in any territory from macro, meso and micro levels. Moreover, Achiquen Millán et al. (2021) consider that these ecosystems have been defined as a set of interconnected and coordinated elements intended to promote entrepreneurship, since universities organize the flow of knowledge to enable the creation of these ecosystems. Even more, the ecosystem according to Pedroza-Zapata and Silva-Flores (2019) is made up of a group of interrelated actors in a given field consisting of at least the following blocks: I+D organization, skilled human resources, formal and informal networks, venture capitalists, professional service providers, and an entrepreneurial culture that connects all this openly and dynamically.

COVID-19 pandemic has impacted the entrepreneurial ecosystem globally. Around 70% of new start-ups have had to terminate their employee contracts, Bennett et al. (2020). Furthermore, out of the three continents, North America experienced the largest employee layoffs at 84%, followed by Europe at 67% and Asia at 59%, Dávila (2020). On the other hand, it also creates opportunities for new products and services due to increased demand. Governments have provided financing to small businesses and new start-ups to support them during this time of crisis, Arundale and Mason (2020).

2.1.1. Components of the Entrepreneurial Ecosystem

According to Isenberg (2010), author of the biological metaphor of ecosystems to understand the entrepreneurial phenomenon, this concept has gained popularity in the academic world of the 21st century, and he proposed that politics (leadership, government), markets (networks, early customers), finances (financial capital), human (educational institutions, labor), culture (success stories, social norms), and supporting capital (infrastructure, supporting professions, non-governmental institutions) combine in complex ways, each stimulating entrepreneurship driving business growth.

Of the previous ones, according to Bóveda et al. (2015) three are essential: innovation, entrepreneurs and financing. Innovation is made up of universities, research centers, laboratories and connections to the outside world. Entrepreneurs are people who have been trained and understand the entrepreneurial spirit and live with the University Center for Technology Transfer. The financing includes financial support from the public sector, as well as seed capital and private equity investments and private availability of angel investments.

In this way, the entrepreneurial ecosystem according to Achiquen Millán et al. (2021) assumes an important role to understand the relationships between the entrepreneurial process and its local environment and is a policy tool to help catalyze a sustainable, entrepreneur-led economy.

According to Spigel and Harrison (2017), the startup ecosystem approach plays an important role in understanding the relationships between the entrepreneurship process and the local environment and is a policy tool to help promote a sustainable and inclusive economy.

According to Cabellero et al. (2014), the recognition that a large part of the Peruvian business movement is concentrated in the capital, a general business ecosystem, has made it possible to identify entrepreneurial companies that operate in the city of Lima. Ruiz et al. (2016) argue that the identification of the Peruvian entrepreneurial ecosystem

stands out for its high level of commitment and low economic sustainability, but with significant participation in the PBI leveraging productivity indexes, well-being and access to social development opportunities for a community that wants to create a business with sustainable development. In agreement Kelley et al. (2016) state that Peru has had a high level of entrepreneurial initiative; almost one in four Peruvians carry out some type of entrepreneurship, due to the large unmet needs that do not cover the needs of self-employment and weak government and private sector support for innovative initiatives and investment ideas. INEI (2021) and amid COVID-19, Peruvian creativity occupied the fourth place with the greatest intention to undertake in Latin America, and the eighth in the world. Likewise, in the department of Lima, 51.4% of entrepreneurial companies are led by women and 48.6% by men.

In this startup ecosystem scenario, there is an opportunity to create new poles of development through competitive and sustainable diversification of production, employment and consumption.

2.1.2. Entrepreneurial Ecosystem in Peru

The study of the Peruvian entrepreneurial ecosystem has not been included in the Global Entrepreneurship Monitor (GEM) study in 2021 and 2022 GEM (2021).

However, if it is, in the Doing Business Ranking that measures economic development in terms of the country's ease with starting new businesses, where it can be observed that New Zealand is in first place with a score of 86.8 points, First place in Latin America is held by Chile (59th place with 72.6 points), followed by Mexico (60th place with 72.4 points) and Colombia (67th place, with a score of 70.1 points), while Peru ranks 76th with a score of 68.7 points. Within this score, the highest indicator is reached by the opening of the business (82.13 points), followed by obtaining permits for construction (72.53 points) and registration of the property of the company 72.11 (points). The lowest score is in compliance with contracts 59.07 points, World Bank (2020).

It is also found that the study The Global Entrepreneurship Network (GEN) occupies the 40th place in the Index of Dynamic Entrepreneurship (IDE), which performs a specific analysis of the relationship between MSMEs, their systemic conditions and entrepreneurship, Kantis et al. (2022).

The body in charge of regulating public policy that favors the entrepreneurial ecosystem is the Ministry of Production, which has created various strategies aimed at promoting entrepreneurship; within these, four stages are identified.

The first one was born on 19 July 2006, when the Government of Peru and the Inter-American Development Bank (IDB) signed Loan Contract No. 1663/OC-PE, giving rise to the FINCyT Science and Technology Program that began in 2007 and had a budget of USD 36 million, and USD 25 million from the IDB; In 2009 the name was changed to FIDECOM and its budget was S/260 million soles.

The second one, in 2013, begins with the FINCyT 2 innovation for competitiveness project with a budget of USD 100 million and USD 35 million from the IDB; Innóvate PERU was created in 2014 and a year later, the PDC-PAC MIPYME fund was incorporated with 50,000 million soles.

The third one, in 2016, began the FINCyT3 Project to Improve Innovation Levels at the national level with a budget of USD 100 million plus USD 40 million from the IDB and this same year FOMITEC was incorporated with 57 millions of soles.

The fourth one, in the year 2021 with the creation of the National Program for Technological Development and Innovation-ProInnóvate through Supreme Decree No. 009-2021-PRODUCE, which also incorporates the Micro Pequeña y Mediana Empresa (MIPYME) [Micro Small and Medium Enterprise] fund that was created by the Law No. 30230, which establishes tax measures, simplification of procedures and permits for the promotion and revitalization of investment in the country with 64.4 million soles and will be administered by the Development Finance Corporation (COFIDE), with this, the Program of Development begins. Innovation and Technological Modernization and Entrepreneurship FINCyT4

with a budget of USD 150 million and USD 100 million from the International Development Bank. PYME (2022); COFIDE (2022); MinPro (2022); ProInnovate (2022).

The Management Committee of the MIPYME Fund is the body in charge of ensuring compliance with the policies, strategies and objectives of the aforementioned fund. It is made up of representatives of the Ministry of Economy and Finance, the Ministry of Production, the Ministry of Agriculture and Irrigation, the Ministry of Foreign Trade and Tourism and the National Competitiveness Council, MinPro (2022); ProInnovate (2022).

The orientation of the investment seeks the development of entrepreneurship, business innovation, productive development, ecosystem institutions, with the development of a series of calls in five categories: (a). innovative ventures with support of up to 50,000 soles, (b). Dynamic ventures up to 140,000 soles, (c). strengthening incubators and accelerators up to 642,900 soles, (d). strengthening networks of angel investors 642,900 soles and (e). the Startup Peru program with up to USD 38,000, ProInnovate (2022).

According to the results of the convocation, the Peruvian entrepreneurial ecosystem currently has 27 recognized incubators, accelerators and angel investors, of which 15% are accelerators and angel investors, 18% are incubators and accelerators and 67% are incubators, also has a Network of Business Support Centers, which are a physical platform of free services aimed at MIPYME and with the National Association PYME Peru, whose main objective is to promote the institutional strengthening of the small business sector. All this is with the aim of strengthening business capacities and achieving a positive impact on the business fabric in PYME (2022); COFIDE (2022); MinPro (2022); ProInnovate (2022).

*2.2. Competitiveness*

According to Gómez (2018), it is the degree to which a country, state, region, or company that produces goods or services faces market competition while increasing the real income of employees and, therefore, their business productivity. Secondly, Castillo et al. (2021) define competitiveness as the company's ability to provide a product or service of the desired quality and follow customer requirements and relevant standards at the lowest possible cost, for Croes et al. (2020); Abreu-Novais et al. (2016), showed that competitiveness is related to the ability of human capital to outperform competitors in terms of feasible economic performance. Nevertheless, Chudnovsky (1991) argues that increasing productivity, especially labor productivity, is a necessary but not sufficient condition to increase competitiveness. Another aspect that has been demonstrated to affect competitiveness is knowledge; as well, Cruz Gonzalez et al. (2013) found that companies that demonstrate higher levels of competition, dynamism, innovation and value creation are those that show greater commitment and activity in learning, thus demonstrating the positive impact of new production factors such as knowledge, efficiency and competitiveness of organizations.

Dieppe (2020) highlights that the World Bank has found that adverse events such as natural disasters, wars and financial crises, or economic shocks such as COVID-19 are associated with large and prolonged drops in productivity. At the current rate, it is estimated that it will take more than a century to close the productivity gap halfway between emerging and advanced economies by half.

According to Dávila (2020) the impact of temporary or permanent closures forced companies to lay off employees or send them on leave without pay, and even slight reductions in the number of employees resulted in reduced competitiveness. Ozili and Arun (2020), during this time, employees are also under psychological stress due to work or family stress, which has an accumulative effect not only on competitiveness but also on productivity during these months, strengthening the entrepreneurial ecosystem to compete in the market.

2.2.1. Business Intelligence and Competitiveness

The continuous monitoring of signals from extreme environments, especially those that allow us to predict future conditions, react to extreme circumstances or act deliberately, possibly thanks to a set of competitive capabilities that companies must implement and

that are to be understood as business intelligence. According to Cubillo (1997), as it is understood, business intelligence is a set of capabilities possessed or mobilized by a business organization to access, collect, interpret and prepare high-value knowledge and information to support the decision-making necessary to design and implement a competitive strategy.

Due to the complex realities of the environment in which entrepreneurship unfolds, business intelligence is not limited to strictly controlling scientific and technical aspects. A comprehensive understanding of strategies enables entrepreneurs to make decisions such as market size, perception of potential projects, regulatory and social frameworks, and the forces of supply and demand that affect business competitiveness is essential. Oña Aldama and de Armas (2015) consider that business intelligence provides the knowledge to define strategies, design I+D programs, cooperation agreements, and implement new technological advances to identify investment and marketing opportunities.

### 2.2.2. Business Intelligence Cycle

For Pineda et al. (2022), business intelligence is more than a set of tools to analyze raw data to help make strategic and operational decisions. It is a framework that provides guidelines to understand that researching different sources of information is a continuous cycle of analysis, knowledge, operations and measurement of business intelligence. In the business intelligence analysis process, management indicators or KPIs (operational indicators) play an important role as a standard for management control. Defining the right KPIs will help you accept and categorize the volume of data and rank it based on its importance to competitiveness. According to Peña Veitía et al. (2019), KPIs are relevant to consider when developing each of the existing business models and achieving strategic goals, and with them the success of the business. Carro Cartaya and Carro Suárez (2008) affirm that entrepreneurial intelligence is a process, a function, and a product.

The business intelligence cycle aligns the objectives with the company's strategy, evaluates the progress of the activities, filters the relevant information and, above all, allows the measurement of the results. In other words, Tello and Velasco (2016) view business intelligence as a set of strategies, actions and tools that focus on managing and creating knowledge from the analysis of existing data in the company.

Therefore, business intelligence is an effective tool to improve business competitiveness and knowledge management. Indeed according to Larson (2009) knowledge management supports the actionable strategies that a smart company can implement to provide a competitive advantage and added value to a product or service because they promote the efficient production and performance of people and that can hardly be replicated by those that do not have plans or defined strategies.

### 2.2.3. Competitive Factors

According to Porter (1991), this model considers six factors related to the supply chain as a factor that allows companies to produce products and services to be competitive in the market. Demand factors are those that allow an organization to know the behavior of the market. According to Vilchez et al. (2022) markets provide valuable insight into consumer behavior, preferences, and customer influences throughout the buying process. Factors related to market opportunities must be related to the organization's ability to see and detect cyclical conditions that can kill profits. Oyaga Martínez et al. (2022) affirms that a company fails if it cannot identify an opportunity, and some experts see opportunities where others only see problems. The opportunities found lead to new opportunities, greater initiative, business growth and inclusion of other available markets. The presence of institutional or public support can have a positive impact on the company in the entrepreneurial ecosystem because it activates work capital and the investment Vera (2021) argues that the implementation of public policies helps to adjust and improve the conditions of competitiveness in the production chain. Another factor is the integration with partner companies which will strengthen the search organization and support its external relations, at various stages of

production. Finally, the elements of the business strategy are factors based on information gathered in the environment that allow the organization to make decisions and structure strategies related to the organization's systems.

### 2.3. Sustainable Development

According to Rengifo Medina et al. (2022), sustainability is one of the most controversial aspects today. It reconciles political, social, educational, legal, and economic factors that conflict with the advance of Eurocentric rationality, the predatory nature of nature, and human dignity. Mejia et al. (2021) Its emergence was determined by the growth of global industrial capitalism, pollution problems and overpopulation in the world. The use of this term is due to the report of the World Conservation Union, which addresses the need to protect the environment, without neglecting the quality of human life.

According to Pernia et al. (2022), sustainable development is an integral part of the perspective pursued with measures to reduce disaster risk and adapt to climate change; it helps reduce the fragility of communities exposed to environmental risk problems, due to displacement as a result of disasters and environmental change, strengthening the capacities of governments and partners to face the challenge of migration with productive policies that promote the sustainable development of economic sectors. According to Mercado (2022), the 2030 Agenda for sustainable development is observed as an action plan that sets out proposals at the global level to achieve development that benefits people, the planet and prosperity. Embedded in the vision of this organization is the desire for world peace, the eradication of all forms of poverty, gender equality, the promotion of effective recognition of the work of women and girls, and the realization of human rights, all of which arises from the perspective of sustainability.

Moreover, García González et al. (2022) hold that sustainability implies the use of an interdisciplinary model that follows new theoretical and conceptual elements in the research and practice of sustainable development, which allows for the understanding and resolution of complex questions that arise between social ecology, which can be very useful in the context of the presence of entrepreneurs looking for opportunities to improve or create new products within the framework of sustainability.

Based on these concepts Alfonso et al. (2016), sustainable development is a development that can meet the needs of present generations without affecting their ability to do so in the future. This definition supports the concept of development under three pillars: social, economic and environmental; This leads to prioritizing the environment and society in development processes that did not exist until now. Even more, Martinez (2022) stresses that if we take into account that, in addition to these distinctive characteristics, these axes are associated with solidarity, the controlled extraction of resources, the management of collective property, commitment to the construction of a different world and respecting the exploitation of means.

### 2.3.1. Factors that Influence Entrepreneurship and Sustainable Development

For Mendez-Picazo et al. (2021), worrying about environmental conditions in many countries has led to a growing interest in taking action to address current problems without affecting future generations. This leads to a rethinking of what is to be achieved, leading to a greater focus on sustainable development, and the emergence of new activities that facilitate the emergence of new economic agents in economic activity, in our case, the entrepreneurial ecosystem.

In this way, sustainable development has become a fundamental objective of political planners. In addition, in light of this new perspective, it is convenient to understand the factors that influence the development of entrepreneurship. Consequently, the contribution of entrepreneurship has emerged as a new factor that should aim to transform economic growth into sustainable development, Johnson and Schaltegger (2019); Schaltegger et al. (2020) so as not to affect the situation of future generations through current policies aimed at achieving current well-being.

It is important to understand the factors that can influence entrepreneurship, as this will help design appropriate measures to promote sustainable development through entrepreneurial activities. In this case, the main factors are classified into two broad categories: socio-cultural and economic.

Sociocultural factors according to Flores et al. (2019): Sociocultural factors favor the management of natural resources; these are acquired through language, observation and practice; reflecting the socio-ecological relationships that occur in the environment. It is this behavior that creates the cultural adaptation of society to nature, which leads to the use of biodiversity. Some social factors are related to the level of education, occupation, gender and age of the person; while culture includes the customs and traditions of a society, which can influence the entrepreneur's creation and take responsibility for sustainable development.

Mendez-Picazo et al. (2021) In sociocultural factors, the importance of the social environment to stimulate entrepreneurial activities and sustainable development should be considered, mainly from two angles: From the perspective of physical institutions and the perspective of institutional structure.

From an institutional perspective, effective institutions are needed to protect property rights, with economic agents more interested in the development of sustainable business practices, with clear rules of the game that do not cause delays in decision-making due to excessive bureaucracy, that do not harm the development of the economic activity of the companies. Thus, Simao and Silveira (2021) consider that the degree of effectiveness and assertion of the actions carried out by government agencies has become the guide in the analysis of any action.

The inconsistency of effective management models, conceptual methods and analyzes in the field of public management still leave much to be desired in terms of the ability of the parties to deliver services equitably to meet the needs of society. Definitely Ferrari Mango (2021), the administrative authorities that interest us are those that intervene in social programs in the field of social policy research and territorial action. They have the potential to act as mediators between the norms that create them and their adaptation to sociocultural interests.

From the perspective of institutional structure, it can be classified into two groups: formal and informal. Formal institutions are characterized by having a very strong cultural component that motivates entrepreneurs to carry out their business projects with a vision of sustainability. In other words, the rules designed by this type of institution to promote economic activity and reduce corruption will have a positive impact on business entrepreneurship, Berdiev and Saunoris (2018); Cherrier et al. (2018); Zhang (2019). In this context, education and skills will make it easier for entrepreneurs to improve and innovate productivity with greater added value, respecting the characteristics of alternative development.

Concerning informal institutions Barozet et al. (2020) consider that in politics they can vary and influence formal decision-making in many ways and levels, either through the creation or application of norms, rules and legislation that benefit private interests, unregulated lobbying, electoral processes. The influence and appointment of positions, without considering meritocracy, allocate public funds to strengthen the sociocultural status of individuals or groups or to limit competition, whether in the executive, legislative or judicial spheres. Moreover, cronyism selects candidates for public and private positions or the granting of benefits from a network of friends; nepotism is not far from this reality and does so through relatives. Either way, the rules of the regime deserve to be broken because the common good is biased.

Economic factors, this second element analyzes various variables that link and directly promote the development of entrepreneurship with sustainable development practices. In the first place, there is the fiscal policy designed by governments, which, within the framework of the type of fiscal policy management used by them, becomes the fundamental tool to achieve macroeconomic balance. This is achieved by stimulating the participation

of companies in the market through their spending policies, improving the behavior of the economic cycle and by assuming a position opposed to the growth of anti-cyclical tax collection, recessive or contractive periods, Portillo (2021).

With higher tax collection, the state is better able to improve income distribution and invest in education and I&D, Mendez-Picazo et al. (2021) components of sociocultural factors. Government measures are also aimed at increasing employment, competitiveness and the preparation of human capital through technological innovation.

Franco and Graña (2020) consider that technology responds endogenously to an increase in the supply of more educated workers by increasing their demand. The factor that unites this relationship is the size of the market; the larger the share of high-quality workers, the more profitable a company's investment in new technology is, as it complements a growing pool of workers. Since these sociocultural and economic factors are interconnected, entrepreneurs will practice sustainable development more responsibly.

### 2.3.2. Elements of Sustainable Development

According to Carro Suárez et al. (2017), sustainability must be based on the interaction of four elements at the organizational level:

1.  Environmental dimension. Considering the prevention of pollution and the rational management of natural resources, recognizing the design of green products from the raw material to the end of the cycle.
2.  Social dimension. Seek benefits for employees and employers through effective human resource management that promotes health, safety, and economic growth, regardless of the company's location.
3.  Economic dimension. It proposes both economic and social benefits, seeking results and benefits from investments in technology to reduce energy consumption and improve the quality of environmental processes, as well as giving something back to the community at a social, economic and environmental level for blow.

## 3. Methodology

The study of the entrepreneurial ecosystem is a complex phenomenon, Roundy et al. (2018) that has not yet been fully presented to the scientific community, especially in Latin America, Lopez and Álvarez (2018). For a better understanding of the problem to be investigated, it is necessary to collect, analyze and process data to present empirical evidence on the importance of the investments made by public policy in the entrepreneurial ecosystem, and for this, the following research question is asked: Is the ecosystem entrepreneur is a significant predictor of competitiveness and sustainable development in the Peruvian case? The answer is developed through a quantitative and explanatory approach.

To achieve the objective of the study, a structural model was proposed (see Figure 1). This model shows the following hypothesis to be evaluated: For the Peruvian case, the entrepreneurial ecosystem is a significant predictor of competitiveness (H1) and of sustainable development (H2).

In order to achieve the objective of this research, data were collected from entrepreneurs, businessmen, university professors, university students, and directors of incubators whose information was provided by the Peruvian Business Association. A search was made for research articles in scientific databases such as Scopus published in the last 5 years in the highest quartiles (Q2, Q3 and Q4), providing information on the variables studied that are important for fieldwork.

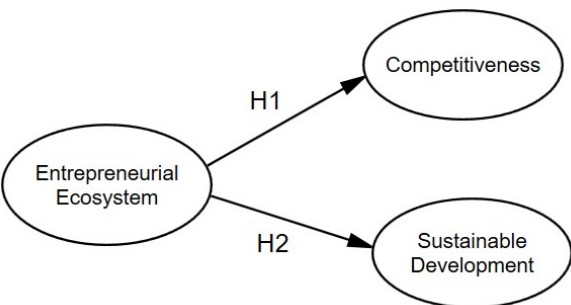

**Figure 1.** Hypothetical model for structural relationship between ecosystem entrepreneur and competitiveness and of sustainable development.

### 3.1. Participants

The participants in this study were entrepreneurs, businessmen, university professors, students and incubator directors who are part of the actors of the Peruvian entrepreneurial ecosystem.

The study was carried out from June to July 2022. The sampling scheme used was non-probabilistic for convenience. The measurement instrument was applied in the Question pro software and its QR code or link was distributed in entrepreneurship conferences scheduled for this purpose and also, the link was shared virtually to a convenience sample through emails and social networks.

The original sample used to test the hypothesis consisted of 240 participants. Eight cases that had more than 10% non-response were removed from the sample. For the 55 cases that showed no response of less than 10%, imputation was used using linear interpolation. Before the statistical analysis, multivariate outliers were identified using Mahalanobis distance, based on the conservative criterion $p < 0.001$ recommended by Kline (2011). Thus, 16 cases with atypical observations were removed, so that the final sample consisted of $n = 216$ valid cases, of which 46% were women and 54% were men.

On the other hand, 65% of the members of the final sample stated that they were between 20 and 30 years old, while 23.3% were between 31 and 40 years old, 9.2% between 41 and 50 years old, 2.4% older than 50 years of age. age. Regarding the participants' line of business, 6.2% corresponded to the industrial sector, 50.5% to the commercial sector, 22.4% to the services sector, while 20.9% indicated that they did not have their own business.

### 3.2. Measuring Instrument

The measurement instrument used in this research was made up of the following scales: (1) Perception of the Entrepreneurial Ecosystem, (2) Perception of Competitiveness and (3) Perception of Sustainable Development. The Entrepreneurial Ecosystem Perception scale, made up of 16 items, was created specifically by the researchers for this study, based on a review of the literature and the conceptual documentation of the construct and its dimensions. In the results section, the factorial structure and the reliability of this scale are discussed.

The competitiveness perception scale, developed by Sarmiento Reyes and Delgado Fernández (2020) contains 24 items distributed in the dimensions (1) client-market, (2) social, (3) technical, (4) economic-financial and (5) environmental. On the other hand, the scale of Perception of Sustainable Development developed by Valencia-Cruzaty (2018), contains 10 items distributed in the dimensions (1) environmental, (2) economic and (3) social.

The instrument, which in the initial stage contained 50 items, was subjected to a content evaluation using a relevance and clarity test by a group of teachers knowledgeable in the area of administration and accounting, as well as a research expert. The content of each item was rated using a scale from zero to five points, where the highest score represented content

with the highest level of clarity, relevance and representation of the construct in question. The items that did not reach an average higher than 3.5 were discarded, in addition to incorporating the suggested improvements to the items by a group of experts. This left a version with 42 items of closed questions under a five-point Likert-type scale, from 1 (totally disagree) to 5 (totally agree).

Next, a pilot test was carried out for this second version of the instrument, which was applied to 25 volunteer participants. Based on the comments of the participants, as well as the judgment of the researchers, 10 items were removed.

As the competitiveness scale of Sarmiento Reyes and Delgado Fernández (2020) and the Sustainable Development of Valencia-Cruzaty (2018) coincide in the economic, social and environmental dimensions; the first had 14 items and the second 10 items. With the theoretical foundation of Sarmiento Reyes and Delgado Fernández (2020) who establish that these dimensions can be used to measure economic, social and environmental development at a regional or national level, the decision is made to select these three dimensions to measure the variable perception of sustainable development. Therefore, a selection and adaptation of 11 items are made to conform to this scale.

Based on the above, for the variable perception of competitiveness, the following dimensions were selected: customer-market, technical and financial, as they are considered as criteria by Secretaría de Economía (2017) to declare a viable and competitive project from the business perspective, an aspect that is confirmed by Sarmiento Reyes et al. (2019).

Subsequently, the 32 items instrument was subjected to a second pilot test. At this stage, the instrument showed acceptable reliability, obtaining a Cronbach's alpha coefficient of 0.938. Based on the above, it was applied to the selected sample. The items and dimensions of the instrument are shown in Table 1.

For the validation of the scales that make up the measurement instrument, exploratory factor analysis (EFA) was used, as well as Cronbach's alpha reliability coefficient, using the statistical package SPSS version 26. To test the hypothesis of this research, a model of structural equations using the Rosseel, Y. Lavaan package Rosseel (2021) of RStudio software version 2022.07.01. The statistical significance level was set at 0.05.

**Table 1.** This is a table caption. Tables should be placed in the main text near to the first time they are cited.

| Dimension No. | Dimension | Item Codes | No. of Items |
| --- | --- | --- | --- |
| 1 | Entrepreneurial ecosystem dimension resources and support practices | EERPA2, EERPA4, EERPA1, EERPA5, EERPA6, EERPA3 | 7 |
| 2 | Entrepreneurial ecosystem business development | EEDN8, EEDN9, EEDN7, EEDN10, EEDN11 | 4 |
| 3 | Technical - financial competitiveness | CTF15, CTF20, CTF17, CTF18, CTF21, CTF19, | 6 |
| 4 | Customer - market competitiveness | CCM14, CCM31, CCM12, CCM13 | 4 |
| 5 | Sustainable social - economic development | DSSE27, DSSE28, DSSE29, DSSE26, DSSE30, DSSE31, DSSE16 | 7 |
| 6 | environmental sustainable development | DSSA25, DSSA24, DSSA23, DSSA22 | 4 |
| TOTAL | | | 32 |

Source: self-made.

## 4. Results

### 4.1. Validation of the Perceived Scale of the Entrepreneurial Ecosystem

The construct validity for the scale of measurement of the perception of the Entrepreneurial Ecosystem was carried out by employing an EFA. According to Hair et al. (2014) for an EFA the sample size should preferably be greater than 100 observations, while

the minimum acceptable sample size is five observations for each variable to be analyzed. However, a widely accepted rule of thumb is a ratio of 10 observations for each variable. In this way, it was considered that the sample size in this research was sufficient to carry out the EFA (Table 2).

For the EFA, principal axis factoring extraction method was used with a Promax oblique rotation, since the factors to be extracted are significantly related to each other ($r = 0.616$). For the identification of the number of factors to be extracted, the Kaiser criterion was used, as well as the scree plot of Catell. The Kaiser-Meyer-Olkin measure (KMO = 0.891) suggested sampling adequacy, while the Bartlett sphericity test was significant ($p < 0.001$), which demonstrated evidence of a sufficient correlation between the variables to carry out the analysis. In this way, two factors were extracted that together explain 64.78% of the total variance. Factor loadings greater than or equal to 0.5 were considered significant, with values for these loadings ranged from 0.597 to 0.940, while the values of the communalities ranged from 0.429 to 0.893.

**Table 2.** Exploratory factorial analysis for the Entrepreneurial Ecosystem Perception scale ($n = 216$).

| Item | EARPA | EEDN | Communalities |
|---|---|---|---|
| EERPA1. Existing external sources of financing for entrepreneurs are sufficient. | **0.655** | 0.403 | 0.429 |
| EERPA2 In your country, the sources of public financing for entrepreneurs support the start-up of businesses. | **0.753** | 0.375 | 0.579 |
| EERPA3. A business incubator or business accelerator can realistically help an entrepreneur obtain international financing. | **0.653** | 0.453 | 0.431 |
| EERPA4. Do you consider that the entrepreneurial ecosystem in your country leads enough activities to promote the entrepreneurial culture? | **0.771** | 0.477 | 0.594 |
| EERPA5. Considers that the business incubation and/or acceleration programs had sufficient impact. | **0.759** | 0.469 | 0.577 |
| EERPA6 It is easy to access external advisory services, accountants, lawyers, and specialists in different areas. | **0.742** | 0.528 | 0.559 |
| EEDN7. Considers the support of the entrepreneurial ecosystem adequate for the start-up of the business. | 0.595 | **0.796** | 0.651 |
| EEDN8. Considers adequate advice in the preparation of the business plan. | 0.508 | **0.940** | 0.893 |
| EEDN9. Considers that the procedures for the development of the business were agile and efficient (register the company, permits, affiliations and others). | 0.674 | **0.597** | 0.507 |
| EEDN10. Consider that the implementation of a business plan allows you to improve your competitiveness. | 0.430 | **0.705** | 0.498 |

Note. Significant factor loadings are shown in bold factors. EERPA: Entrepreneurial ecosystem dimension resources and support practices, EEDN: Entrepreneurial ecosystem dimension business development. Source: self-made.

The first factor grouped six items, as shown in Table 2, which together explain 52% of the total variance. This factor was called Resources and Support Practices, which includes questions that have to do with financing entrepreneurs and questions about public policy strategies to support entrepreneurs.

The second factor grouped four items that have to do with the support for the start-up of the business, the business plan and the procedures necessary to operate; together they explain 12.78% of the total variance. This factor was called Business Development.

For some items, significant factorial loads were observed in both factors, however, the item was assigned to the factor that contributed the greatest factorial load, except for the EEDN9 item, where despite a lower load, it was decided to keep the item in the Development factor of the Business, because within the intention of the investigation, it is intended to evaluate whether the entrepreneurs carry out the procedures to legalize the company.

However, it is understood that the question has a higher factorial load in the EERPA factor under the conceptual contribution, since the theoretical postulate of the Global Entrepreneurship Monitor (GEM) study, within the framework of national conditions for entrepreneurial activity, describes a series of factors that influence business development

with nine indicators, and in indicator number two it contemplates government policies (support and relevance, as well as taxes and bureaucracy), Quezada et al. (2019). In addition, as established in the index Doing Business, which covers 12 areas of business regulation, including procedures to start a business, obtain permits, registration, taxes and others, World Bank (2020).

### 4.2. Validation of the Scales for Perception of Competitiveness and Perception of Sustainable Development

In the same way as in the validation of the Perception of the Entrepreneurial Ecosystem scale, for the construct validation of the Perception of Competitiveness and Perception of Sustainable Development scales, an EFA was used using principal axis factoring and Promax rotation. The number of factors to extract was identified using the scree plot and the Kaiser criterion. In this way, for the Competitiveness Perception scale, two factors were extracted that together explain 62.50% of the total variance (see Table 3). Similarly, for the Perception of Sustainable Development scale, two factors were extracted that together explain 67.84% of the total variance (see Table 4). For both scales, evidence of sample adequacy was obtained (*KMO* = 0.935 for Competitiveness, *KMO* = 0.925 for Perception of Sustainable Development) and of sufficient correlation between the variables (Bartlett test $p < 0.001$ for both scales).

**Table 3.** Exploratory factorial analysis for Competitiveness Perception ($n = 216$).

| The Scale of Perception of the Competitiveness | MCC | CTF | Communalities |
|---|---|---|---|
| CTF11. The increase in the supply of goods and services was constant according to the behavior of the market. | 0.651 | **0.701** | 0.518 |
| CTF12. The number of suppliers regularly supplied stock of materials to start their business. | 0.614 | **0.758** | 0.575 |
| CTF13. Identifies loyalty by keeping track of the number of customers who demand the products or services of your business. | 0.648 | **0.737** | 0.555 |
| CTF14. You are satisfied with the quality and price of the product offered by your business. | 0.598 | **0.795** | 0.634 |
| CCM15. The liquidity of the business could meet its business obligations in the short term. | **0.682** | 0.621 | 0.484 |
| CCM16. The investments of the business venture generated greater value or economic growth. | **0.775** | 0.629 | 0.602 |
| CCM17. The productivity of the goods and services produced by each factor of production (labor, capital, natural resources and technology) were used optimally. | **0.787** | 0.576 | 0.624 |
| CCM18. The areas that generate income from innovations were economically profitable. | **0.698** | 0.596 | 0.493 |
| CCM19. Intellectual property concerning creation and innovation allowed him to obtain recognition and profits. | **0.694** | 0.644 | 0.507 |
| CCM20. The evaluation of the obsolescence of technological equipment was planned based on the life cycle and production levels. | **0.749** | 0.632 | 0.566 |
| CCM21. Having information technology made it possible to obtain quality information. | **0.666** | 0.574 | 0.451 |

Note. Significant factor loadings are shown in bold. Factors. CCM: Competitiveness client-market dimension, CTF: Technical-financial competitiveness. Source: self-made.

The reliability of the Entrepreneurial Ecosystem Perception scale was evaluated using Cronbach's alpha coefficient. For the complete scale, a value of $\alpha = 0.897$ was obtained, while for the Resources and Support Practices factor $\alpha = 0.867$ was obtained and for the Business Development factor $\alpha = 0.834$ was obtained, which according to Hair et al. (2014) and Taber (2018) suggest an adequate degree of consistency between the measurements of the variable.

**Table 4.** Exploratory factorial analysis for Sustainable Development Perception (*n* = 216).

| Sustainable Development Scale | DSA | DSSE | Communalities |
|---|---|---|---|
| DSI KNOW22. I consider that the environment is respected in the development of the activities of your business. | 0.573 | **0.781** | 0.610 |
| DSI KNOW23. The control of the waste management program produced by your undertaking protected health and the environment. | 0.654 | **0.826** | 0.687 |
| DSI KNOW24. Complied with the regulations of respect for biodiversity in the development of its enterprise. | 0.624 | **0.805** | 0.650 |
| DSI KNOW25. The recycling policy, from their undertaking turned into new products or into material resources with which they manufactured other products. | 0.585 | **0.791** | 0.626 |
| DSSA26. Human resources took care of selecting, recruiting and hiring people with ease. | **0.734** | 0.553 | 0.538 |
| DSSA27. Attended to the payment of wages and the contribution to the social security of its workers. | **0.705** | 0.538 | 0.497 |
| DSSA28. The undertaking of your business supports community development. | **0.771** | 0.636 | 0.604 |
| DSSA29. Your business benefits the state with its taxes and the family with income. | **0.808** | 0.604 | 0.654 |
| DSSA30. Do you think there is an increase in sales due to your business? | **0.807** | 0.612 | 0.652 |
| DSSA31. The presence of your enterprise created poles of development in the entrepreneurial ecosystem. | **0.821** | 0.612 | 0.675 |
| DSSA32. The participation of your enterprise in the market was with its financing. | **0.711** | 0.535 | 0.505 |

Note. Significant factor loadings are shown in bold. Factors. DSA: Sustainable development environmental dimension, DSSE: Sustainable development social-economic dimension. Source: self-made.

For the perception of competitiveness scale, the first factor added seven items, made up of CCM15-CCM21, which together explain 55.23% of the total variance of the client-market Competitiveness factor regarding investment, innovation, liquidity and provision of information technology. The second factor grouped four items CTF11-CTF14 that explain 7.27% of the total variance of the supply of goods and services according to consumer tastes and preferences, supplier logistics, customer loyalty and satisfaction.

On the other hand, the scale of perception of sustainable development, the first factor grouped seven items, made up of DSSA16-DSSA32 that together explain 58.18% of the total variance of environmental sustainable development regarding the selection and recruitment of employees, payment of salaries, business entrepreneurship, business financing and sales. The second factor added to four DSSE22-DSSE25 items that together explain 9.66% of the total variance of respect for the environment, respect for the biodiversity standards of the waste program and recycling policy of the entrepreneurial ecosystem.

The reliability of both scales was evaluated using Cronbach's alpha coefficient. The value of the Perception of Competitiveness scale was $\alpha = 0.919$, while for the corresponding dimension $\alpha = 0.886$ (customer-market) and $\alpha = 0.837$ (technical-financial). Consequently, the value of the entire Perception of Sustainable Development scale was $\alpha = 0.927$, for the environmental dimension $\alpha = 0.907$ and the socioeconomic dimension $\alpha = 0.877$. These results indicate the degree of consistency between the measurements of the variables.

*4.3. Hypothesis Testing*

To test the proposed hypothesis, a structural equation model was used with the estimation of parameters by Maximum Likelihood, which is shown in Figure 2. The sample size used in this study (*n* = 216) was considered adequate for modeling with equations structural Hair et al. (2014). Previously, the fit of the measurement model associated with the structural model was evaluated.

For both models, parcels were used as indicator variables of the latent variables. For the formation of parcels, the average of three or four items was used and selected randomly in order for three parcels per construct to be obtained. According to Matsunaga (2008), the use of parcels in a structural model helps reduce the complexity of the model and mitigate problems associated with the lack of normality, while increasing the efficiency of the model.

The assumption of multivariate normality was evaluated using the Rosyton test. Evidence of violation of this assumption was found, so that the estimation of parameters

was carried out by Maximum Likelihood with robust standard error and scaled Chi-square test of Satorra and Bentler (1994).

Evidence of a good fit was found for the measurement model ($x^2 = 5.331$, $df = 6$, $p = 0.502$). According to Kline (2011) the Chi-square statistic is the most basic statistic for evaluating the fit of the population covariances and the covariances predicted by the model. When this statistic is not significant, there is evidence of a good fit for the model.

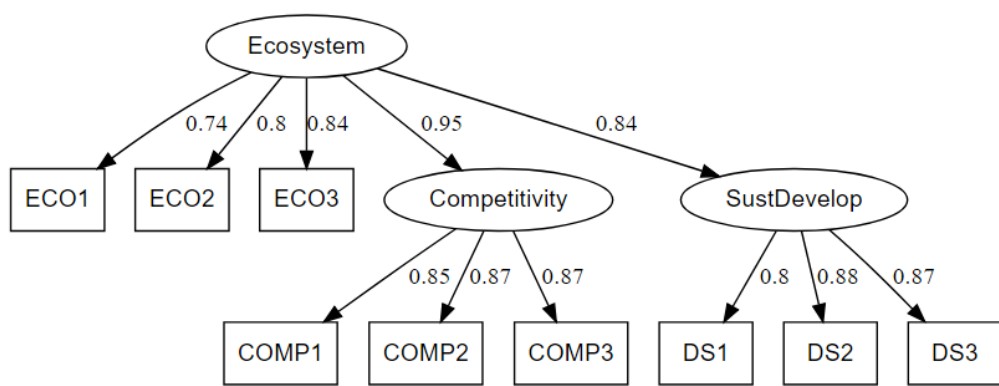

**Figure 2.** Structural model fitted with standardized regression coefficients. Constructs. SustDevelop: Sustainable Development, Competitivity: Competitiveness, Ecosystem: Entrepreneurial Ecosystem.

On the other hand, for the structural model shown in Figure 2, it was found that the Chi-square statistic was significant. However, based on the approximate fit statistics, and using as reference thresholds those suggested by Hair et al. (2014); Hu and Bentler (1999) and Tabachnick et al. (2013), evidence was obtained that the fit of the model was acceptable (Table 5), which validates the hypothesis proposed in this study. In addition, evidence was found that the perception of the entrepreneurial ecosystem explains a significant proportion of the variance of the perception of competitiveness ($r^2 = 0.895$) and the perception of sustainable development ($r^2 = 0.708$).

**Table 5.** Fit statistics for the structural model.

| $x^2{}_{SB}$, | DF | p | CFI | TLI | SRMR | RMSEA | 90% IC |
|---|---|---|---|---|---|---|---|
| 57.132 | 24 | <0.001 | 0.972 | 0.957 | 0.012 | 0.081 | [0.057, 0.105] |
| adjustment threshold | | | ≥0.95 | ≥0.95 | ≤0.08 | ≤0.08 | |

Fit statistics. $x^2{}_{SB}$: Satorra-Bentler scaled chi-square, *df* = degrees of freedom, *CFI*: Comparative Fit Index, *TLI*: Tucker-Lewis Index, *SRMR*: Standardized Root Mean Square Residual, *RMSEA*: Steiger–Lind Root Mean Square Error of Approximation. 90% CI: 90% confidence interval for the *RMSEA*. Source: self-made.

## 5. Discussion

The results of this study suggest there is evidence that the Entrepreneurial Ecosystem has an effect on the level of sustainable development, while it explains an important proportion of this construct. This result agrees with Mendez-Picazo et al. (2021) who found a positive relationship between entrepreneurship in the case of general enterprises as a predictor of sustainable development with a ($r^2 = 0.730$ and significance of $p \leq 1\%$).

To measure entrepreneurship, activities aimed at promoting entrepreneurship through the public sector were observed in two dimensions: the first economic factor and the second sociocultural factor with social impact and policies aimed at redistributing innovations and anti-corruption subsidies, making markets are freer and more efficient.

They used three indicators to measure the relationship between entrepreneurship and sustainable development: (a) Adjusted net national income, which is calculated as gross national income minus consumption of fixed capital and depletion of natural resources (AI),

(b) The United Nations Human Development Index (HDI), measures three dimensions: health, education and a decent standard of living, and (c) The PIBpc variable is the PBI per capita in constant 2010 World Bank dollars (GDPpc). According to the conclusion of Valenzuela-Keller et al. (2021) public and private programs that favor the exposure of young people to entrepreneurship increase their entrepreneurial motivation and this support can also become a driving factor for economic and social development, as confirmed by various studies, saying that entrepreneurship and education are directly related to people with development, Pacheco-Ruiz et al. (2022).

In this form, Soria-Barreto et al. (2021) confirmed the importance of the development of an entrepreneurial ecosystem, which studied the start-up stages of entrepreneurship in Chile and Colombia, explained by three variables: (a) fear of failure; (b) meet entrepreneurs and have networks; and (c) have skills to develop a new business. The last variable, which refers to the development of the capacities and competencies of the company, is the elderly impact. For this, they recommend that to encourage progressive entrepreneurship over the years, the countries must include in their public policies the development of business-friendly ecosystems for encouraging new businesses to grow in terms of innovation and sustainability.

Regarding the entrepreneurial ecosystem and competitiveness, the variance of the level of competitiveness ($r^2 = 0.895$), emphasizes that entrepreneurship must be recognized as a source of development and competitiveness that can positively influence the economic dynamics of societies, Toca-Torres (2010). In addition, maximizing productivity and improve the profile of each employee in each organization, Zárate (2013). Further, Prado et al. (2019), mention that emerging entrepreneurship is not always carried out with the necessary competitiveness or expansion intention. For entrepreneurship to impact productivity, the government must motivate and train entrepreneurs to enter globalized markets with greater competitive advantage and innovations in new products and production processes.

Finally, two important limitations of this study were the following: (1) we investigated only the entrepreneurial ecosystem located in the eastern area of Lima, and (2) the survey used as the main data collection technique was a structured questionnaire that does not allow commenting or giving entrepreneur improvement ideas. In addition, because data were obtained from a convenience sampling scheme, the results of this study cannot be generalized. Nevertheless, the findings of this research can contribute to a better understanding of the relationship between the entrepreneurial ecosystem, competitiveness, and sustainable development in the Peruvian case.

## 6. Conclusions

The determinants of the entrepreneurial ecosystem, competitiveness, and sustainability vary according to the national context, the type of business, the geographical region and the social, economic and political conditions.

Private financing sources and public policies determine the productivity and sustainability of the entrepreneurial ecosystem, so effective programs and strategies can be developed that help strengthen the ecosystem and participate in the market with a greater competitive margin.

The results obtained in this study help to explain the development of literature on theories and concepts that influence the productivity and sustainability of enterprises, and also act as an important tool in decision-making for ecosystem actors.

Due to the aforementioned, entrepreneurship cannot be limited solely to the private business sector, but rather, its scope must be expanded and even projected from the practice of public policies, considering that its social effects can be emphasized and thus expand the positive results with greater socioeconomic coverage.

A structural equation model was used to evaluate the effect of the perception of the entrepreneurial ecosystem on the perception of competitiveness and sustainable development. Before evaluating the fit of the model, content and construct validation of the modified measurement scales from the original versions were performed. Content vali-

dation was performed by a panel of experts in the field, while construct validation was performed through exploratory factor analysis. In addition, the reliability of each scale and its dimensions was evaluated based on Cronbach's alpha coefficient.

Based on these procedures, evidence was obtained that the scales that measure the constructs of perception of the Entrepreneurial Ecosystem, perception of Competitiveness and perception of Sustainable Development showed sufficient content validity, construct validity and reliability. It should be noted that confirmatory factor analysis (CFA) was not performed because a second sample was not used.

In addition, the statistical analysis found that both the measurement model and the corresponding structural model showed a good fit. Which means that there is sufficient evidence to suggest that there is a significant effect of the perception of the entrepreneurial ecosystem that explains a significant amount of the variance of Competitiveness and sustainable development. Despite that in this study a convenience sample was used a convenience sample, it is possible that these findings could be applicable for entrepreneurship in the eastern cone of Lima Peru.

Due to the aforementioned, entrepreneurship cannot be limited only to the private business sector, but rather its scope must be expanded and even projected from the practice of public policies, considering that its social effects can be emphasized and thereby expand the positive results with greater socioeconomic coverage.

## 7. Recommendations

The strengthening of the entrepreneurial ecosystem is a topic of vital importance in the processes of promoting incubators, accelerators, angel investor networks and business development centers in order to impact the promotion of entrepreneurial culture, the formation of entrepreneurs and the creation of companies, so that countries can face the challenges of economic growth that are presented today.

That is why the information obtained with the application of the measurement scale presented in this research becomes an input to be analyzed with a structural equations model, whose results allow us to visualize the degree to which the entrepreneurial ecosystem is a significant predictor of competitiveness and sustainable development.

Based on this information, governments, public and private institutions are recommended to measure their entrepreneurial ecosystem to generate strategies that allow them to develop plans and programs to identify sources of competitive advantage and work in areas of opportunity. It is understood that the scale was built for the Peruvian case, but it can be adapted to different contexts.

## 8. Implications

The strengthening of the entrepreneurial ecosystem favors the development of better business plans, an aspect that is considered very important, since it is a tool that allows the allocation of production factors (capital, labor and natural resources). Thus, it consolidates the efforts to activate entrepreneurship in the eastern cone of Lima, by providing actions that serve as a basis to strengthen administrative, productive, financial, and market processes, without affecting their worldview.

The consolidation of administrative and organizational management will contribute to sustainable conditions for productive activity, since it will be possible to monitor and control the various processes that stimulate work in their environment. Think of entrepreneurship as an initiative that promotes employment, the financial stability of companies and multiplier effects in different economic sectors, which ultimately impact the state, company, and family.

**Author Contributions:** Conceptualization, J.F.B.A.; methodology, J.F.B.A. and A.R.B.; software, A.R.B. and R.R.A.; validation, J.F.B.A., A.R.B. and R.R.A.; formal analysis, R.R.A. and A.R.B.; investigation, J.F.B.A. and B.E.A.; resources, J.F.B.A.; data curation, R.R.A. and A.R.B.; writing—original draft preparation, J.F.B.A. and A.R.B.; writing—review and editing, J.F.B.A., A.R.B. and R.R.A.; visualization;

supervision, J.F.B.A.; project administration, J.F.B.A.; funding acquisition, J.F.B.A. All authors have read and agreed to the published version of the manuscript.

**Funding:** This research was funded by Universidad Peruana Unión and Universidad de Montemorelos.

**Institutional Review Board Statement:** Resolución de la Universidad Peruana Unión N° 0243-2022/UPeU-FCE-CF.

**Informed Consent Statement:** Not applicable.

**Data Availability Statement:** Not applicable.

**Conflicts of Interest:** The authors declare no conflict of interest.

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
