# Peer review of "The Entrepreneurial Ecosystem in the Eastern Cone of Lima as a Predictor of Competitiveness and Sustainable Development"

_admsci, doi:10.3390/admsci13010012_

Round 1

Reviewer 1 Report

Thanks for providing me with this precious opportunity to review this paper. I found the work to be clearly written and interesting. The research question which was raised in this work is also appealing. I applaud the authors for developing their own scale. I have some concerns about why authors emphasize COVID-19, as their model can work at any time, no matter whether those times are turbulent or not.

Minor issues

1.     Please check the format of your references. Keep the format consistent

2.     “79.1% of whom were entrepreneurs 11 and businessmen, 6.2% from the industrial sector, 50.5% from the commercial sector and 22.4% from 12 the service sector; the remaining 21% of the sample corresponds to university professors, university 13 students, directors and advisors of incubators and accelerators.” So the majority of professors are entrepreneurs?

3.     Spacing in the manuscript needs formatting

Reviewer 2 Report

General overview:

The manuscript entitled The entrepreneurial ecosystem in the eastern cone of Lima as a  predictor of competitiveness and sustainable development presents an interesting topic, the subject considered by the authors in being certainly one of increasing importance in the current context.

The title attracts the reader’s attention and reflects the actual content of the article, while also comprising key terms that could be easily found during an intelligent search.

To a large extent, the research is correctly structured, following a natural flow and presenting key aspects undoubtedly targeted by the audience. Moreover, the results obtained arouse the attention and interest of the audience. At the same time, the usage of tables and figures is useful in presenting the analysis' main findings. 

Recommendations:

- Within the introduction, several relatively recent research were mentioned. The exposure of the main research question in the introductory part of the paper highlights the points of interest of the authors’ work, this part being extremely helpful for the audience. However, the overall introduction could be improved.

- Figure 4 is mentioned twice in the body of the text, but it cannot be identified visually. Is it possible that the figures are not numbered correctly? Also, please follow the recommendation of the journal, namely: All Figures, Schemes and Tables should be inserted into the main text close to their first citation and must be numbered following their number of appearance (Figure 1, Scheme I, Figure 2, Scheme II, Table 1, etc.).

- It is recommended to verify the spacing between the rows, as well as the font size used, as this must be uniform throughout the entire paper.

- If possible, the authors should also provide a concise viewpoint related to the limitations of the current study. Awareness of the limitations of research is a strong point of it, as it lays the foundations for any possible future work.

- Several discrepancies were observed between the present paper and required template of the journal. Therefore, it would be advisable for the author/s to review the structure and the design of the specific parts within the manuscript (e.g. some of the references format etc.).

Reviewer 3 Report

Thank you for the chance to review the article, I note the following comments:

1.     The introduction needs further improvements, and more literature reviews and empirical studies are needed.

2.     The study problem needs to be clearly mentioned and supported either with empirical studies or with governmental reports. There is a need to discuss the entrepreneurial ecosystem more clearly here and talk about its various components. There are only two references in the introduction section. This is not accepted.

3.     You need to include the questions of the study, the objectives of the study, the research gap, and the organization of the study in the introduction section.

4.     There are no implications of the study.

5.     Authors need to develop hypotheses after every section of the study variables even if he uses only correlation or chi-square.

6.     Conclusion needs to summarize the entire article, you do not need to talk about any technical details here, and any findings technically should be mentioned in the result section not here.

7.     There are no limitations to the study and future studies suggestions.

8.     There are no implications of the study.

9.     The convenience sampling technique should be acknowledged as a limitation of the study due to the bias that is generated.

10.Competitive factor needs more theoretical and empirical support. It is not possible you depend only on one study.

11. Please proofread the article.

-        Please include the following studies and enrich your literature: COVID-19 as an external enabler: The role of entrepreneurial self-efficacy and entrepreneurial orientation (2022).

-        Investigating the impact of institutions on small business creation among Saudi entrepreneurs (2022).

-        Exploring the influence of potential entrepreneurs’ personality traits on small venture creation: the case of Saudi Arabia (2022).

-        Psychological features and entrepreneurial intention among Saudi small entrepreneurs during adverse times (2022).

12. There is a need for more theories and previous studies that support the claims you have.

Round 2

Reviewer 3 Report

Thank you for sending me the article to review, in fact, the author has tried to improve the article however, I still need to report the following:

1.      Abstract still needs to be rewritten with a clear focus on the objective, sample and findings of the study. The author does not need to include any technical details in the abstract section such as RMSEA or others.

2.      The introduction is still poorly written, it hardly contains 3 references, and no clear research problem or context of the study. How is it possible you have only half a page as an introduction with very limited references? The introduction is the area where you discuss the topic in general, then move to your variables and explain them and link them with each other and then talk about the problem of the statement and other things,

3.      As I can see, the sequence of in-text citations is not in sequence and is not according to the journal style, for example, you cannot start with 2 and then move to 66, you should be consistent, starting from 1 and moving to 2.3 and so on.

4.      The responses offered by the authors are not concise and clear, it seems like the author has just copied and pasted from the article, and the improvements are not clear in the newly submitted version. The author seems to ignore the reviewer's comments.

5.      Business intelligence systems and cycles need more support and evidence. Only two references cannot support your arguments.

6.      In line 407 you talked about the hypothesis of the study and so far I have not seen even a single hypothesis in the study. Have you really assumed them? Where are they? Is it a quantitative or qualitative study? I am confused. Figure 2 shows it is a quantitative study, then which means you should have 5 hypotheses.

7.      I already mentioned earlier that if you have used convenience sampling as your sample then should be acknowledged in the limitation as a limitation of the study.

8.      The author seems to only repeat the findings in the conclusion section, it is not clear and concise and it does not summarize the whole article.

9.      I believe that authors should be more serious while dealing with the reviewers' comments, the purpose of reviewing an article is to help in improving its outcome.

Author Response

Good Morning!

I am sending the corrections according to your observations. Attachment

best regards

Round 3

Reviewer 3 Report

can be accepted.